# Minimally Invasive AC Joint Reconstruction System (MINAR^®^) in Modified Triple-Button Technique for the Treatment of Acute AC Joint Dislocation

**DOI:** 10.3390/jcm8101683

**Published:** 2019-10-15

**Authors:** Robert Breuer, Alexandra Unterrainer, Micha Komjati, Thomas M. Tiefenboeck, Klemens Trieb, Christof Pirkl

**Affiliations:** 1Department of Orthopedics and Trauma Surgery, Medical University of Vienna, 1090 Vienna, Austria; 2Department of Orthopedics and Trauma Surgery, Klinikum Wels-Grieskirchen, 4600 Wels, Austria; 3Department of Orthopedics, Herz-Jesu Hospital, 1030 Vienna, Austria; 4Computed Tomography Research Group, University of Applied Sciences Upper Austria, 4600 Wels, Austria

**Keywords:** acromioclavicular joint dislocation, MINAR^®^, minimally invasive, modified technique, horizontal instability

## Abstract

Acute acromioclavicular (AC) joint dislocation is a frequent sports injury with more than 100 different operation methods described. A total of 65 patients with an acute AC joint dislocation were treated with the modified MINAR^®^ system between 2009 and 2013. Clinical outcome, horizontal and vertical instability, as well as concomitant intraarticular injuries were assessed. We used Zanca, stress and axial X-rays for radiological assessment. A Constant score of 95 (±8.8), University of California Los Angeles Shoulder score (UCLA) of 31 (±4.9), Disabilities of Arm, Shoulder and Hand (DASH) of 9.1 (±14.3), and Visual Analogue Scale (VAS) of 0.9 (±0.126) was found. A total of 30 patients (59%) had no signs of reduction loss, nine patients (18%) a slight loss, 11 patients (22%) a partial loss, and one patient (2%) a total loss. No significant influence on the clinical scores could be shown. The postoperative coracoclavicular (CC) distance negatively affected the Constant (*p* = 0.007) and UCLA scores (*p* = 0.035). A longer time interval to surgery had a negative influence on all scores (*p* ≤ 0.001). We could not find any signs of persistent horizontal instability or intraarticular injuries at follow-up. The MINAR^®^ system promises satisfactory functional and radiological results. When setting the correct indication, patients benefit from an early operation. No persisting horizontal instability was observed following suturing of the AC capsule and the delta fascia.

## 1. Introduction

Acromioclavicular (AC) joint injuries account for about 9% of all shoulder girdle injuries [1]. Most occur in male adults in their 20s and are often related to sports injuries [2]. There seems to be general consensus that Rockwood I and II injuries can be treated conservatively with good results [3,4,5,6], even when considering the long-term outcome [4]. Although there is no controversy over Rockwood IV–VI injuries being treated operatively, there is no clear evidence regarding Rockwood III lesions [7]. On this controversial topic some studies point out very good outcomes after conservative treatment, even on elite athletes’ levels [8], whereas others suggest better results with the surgical approach [9]. To facilitate clinical decision making, the International Society of Arthroscopy, Knee Surgery and Orthopaedic Sports Medicine (ISAKOS) upper extremity committee published a consensus statement to aid the subclassification of the solely radiologically diagnosed Rockwood III injuries into stable (IIIa) and unstable (IIIb) [10]. They recommend the surgical treatment of patients with persisting pain and loss of function, which are likely to occur in Type IIIb injuries [10]. In cases with persistent symptoms lasting more than 3 weeks, an Alexander view X-ray is recommended [10]. Posterior dislocations should be classified as IIIb and treated surgically, all other cases are defined as IIIa and followed up after 6 weeks [10]. For radiological assessment of dynamic horizontal instability, Tauber et al., introduced an axillary view [11]. Zumstein et al. recently defined radiological parameters which can be measured in the Alexander view in order to properly assess the horizontal displacement and help to classify these injuries [12]. More than 100 surgical methods have so far been described for acute injuries, but none have been established as a gold standard [13]. Arthroscopic and minimally invasive less rigid methods are on the advance and have generally replaced older techniques including K-wire fixation and the Bosworth screw [5]. Currently under discussion is the problem of persisting horizontal instability after solitary coracoclavicular (CC) stabilization which leads to the recommendation of addressing the AC joint directly [14,15,16,17]. Further controversy exists about the necessity of an arthroscopic approach due to a certain rate of concomitant injuries in cases of acute AC joint injuries [18,19,20].

The aim of our study was the retrospective analysis of the functional and radiological mid-term results of another minimally invasive method for reduction and retention of the acute AC joint dislocation types Rockwood III–V with coracoclavicular reconstruction using a Flip-Button technique (MINAR^®^, Karl Storz, Tuttlingen, Germany).

## 2. Experimental Section

### 2.1. Patients

In this retrospective case series, a consecutive series of 65 patients (62 male, three female) treated with the Minimal Invasive AC Joint Reconstruction System (MINAR^®^, Karl Storz) after an acute AC joint dislocation (time from injury to surgery <21 days [21]) type Rockwood III–V between 2009 and 2013 were included. Every patient was thoroughly informed about surgery, possible complications, and the follow-up treatment. The decision for surgery was made individually for each patient, considering all significant variables including the Rockwood grade, physical demand of the patient, age, and trauma mechanism. Written, informed consent for surgery and participation in our study was obtained and the local ethics committee approved of the trial (Ethics committee of Upper Austria, Austria, vote B-113-16). At the time of follow-up, 51 patients voluntarily participated (21.5% lost to FU follow-up). An overview of the patient demographics is outlined in Table 1.

### 2.2. X-ray

Each patient had X-rays taken in two planes, firstly, AP (antero-posterior, Zanca view) and secondly, an AC-stress radiograph with added weight (5 kg). Horizontal instability was determined clinically and with an axial radiograph, which resembles the standardized protocol at our clinic. Intraoperatively, dynamic axillary views were shot whilst manually trying to provoke AP translation of the lateral clavicle. Postoperatively, the same radiographic series were conducted. At follow-up, Zanca and stress views were taken of both sides for comparison. The AC and CC distances were measured using a reference sphere. Calcifications of the CC-ligaments and the AC joint capsule were documented as well as arthritic changes of the AC joint. All findings were compared pre- and post-operatively (Figure 1).

### 2.3. Clinical Examination and Scores

The clinical examination included rotator-cuff (RC) tests (Jobe Test [22], Champagne Toast Test [23], External Rotation Strength Test [24], Patte Test [25], Lift off test [26], Belly Press Test [27], Bear Hug Test [28]), SLAP (O’Brien-Test [29], Supine-Flexion-Resistance-Test [30]) and biceps tendon tests (Yergason Test [31], Speed’s Test [32]), impingement tests (Neer’s test [33], Hawkins Kennedy Test [34]), a clinical evaluation of horizontal and vertical instability by palpation and manipulation as well as range of motion (ROM) testing in a standardized manner. After surgical treatment, horizontal instability was tested for again by manipulation under direct visualization of the AC joint. Instability was defined as a horizontal translation of the lateral clavicle by more than 50% shaft width. For functional analysis, the Constant- [35] (not age adapted), Disabilities of Arm, Shoulder and Hand (DASH) score [36] and University of California Los Angeles Shoulder score (UCLA) [37] scores were used. Additionally, the patients’ pain was measured using the Visual Analogue Scale (VAS) score. The Constant score ranges from 0 (no function with considerable pain) to 100 (normal function), the DASH from 100 (total impairment) to 0 (no impairment), the UCLA from 0 (no function, no satisfaction) to 35 (normal function, very satisfied). The VAS is a well-known tool for assessment of patients’ pain ranging from 10 (severe pain) to 0 (no pain).

### 2.4. Statistics

A regression analysis was used to examine the relationship between patient age and clinical scores, as well as an ANOVA to analyze differences between clinical score means. Possible differences between the groups were calculated for all clinical outcomes and the loss of reduction with an ANOVA. Further analysis to explore differences between group means while controlling alpha error was carried out with the Bonferroni and Turkey post-hoc tests. A correlation between time to surgery, clinical scores, and the CC distance was determined using the Pearson correlation. In order to determine an effect on the clinical scores in patients operated in less than 10 days after trauma, the Mann–Whitney-U test was performed, respectively the *t*-test was undertaken to determine an influence on CC distance. Different groups of the primary Rockwood grade as well as CC calcification and the clinical outcome were compared using the Kruskal–Wallis test. The same test was used to explore group differences on the effect of surgery in under 10 days and the clinical outcome. The level of statistical significance was set at *p* ≤ 0.05. Statistical analysis was performed using SPSS 22.0 (SPSS Inc., Chicago, IL, USA)

### 2.5. Surgical Technique and Follow-Up Treatment

The patient is placed in a beach-chair position, and the landmarks are marked (Figure 3A). A 3–4 cm skin incision is made over the coracoid (Figure 3B) and the bone is exposed (Figure 3C). Drill holes are made into the coracoid base and the lateral clavicle (Figure 2A). In our modified technique, three flip-buttons (Fliptack^®^, Karl Storz) were augmented with non-absorbable sutures and a zip loop construct is created (Figure 2B). One of the three buttons can be carried through the coracoid, the remaining buttons through the clavicle to create a V-shaped construct and therefore two divergent force vectors (Figure 3D). Interposed discus and capsule parts are removed, and the reduction is performed. We recommend a little over-reduction to counter postoperative loss of reduction. As a technical note we recommend knotting between clavicle and coracoid to avoid irritating sutures directly under the skin (Figure 2B). Next the AC capsule is sutured via crossing transosseous sutures. The deltoid fascia is sutured before wound closure. A wound drain is not deemed necessary. Postoperatively the patient’s arm is immobilized with a sling bandage for 4 weeks. We allow passive and active assisted physiotherapy with up to 90° flexion and abduction during the first four postoperative weeks. After 4 weeks the bandage is removed, and the ROM should be increased in line with the pain threshold. Physiotherapy should be carried out without resistance for 12 weeks. Overhead work and strain over 3 kg should be avoided for 3 months and contact sports for up to 5 months. Removal of the surgical implants is generally not necessary.

## 3. Results

Between 2009 and 2013, 65 patients (62 males, three female) were treated with the MINAR^®^ system following acute AC joint dislocations. Fifty-one (50 males, one female) patients were able to participate in the follow-up (21.5% lost to FU). One patient had to be excluded because of revision surgery. The other patients lost to follow-up either refused to participate or could not be traced. At the time of injury, the median age was 43 years and ranged from 19 to 63 years. At follow-up the mean age was 48 years (22–68). In 30 patients (59%) the shoulder of the dominant arm was affected. Rockwood type V was the main reason for surgery in 32 patients (63%), followed by 11 patients (21%) with Type IV injuries and only eight patients (16%) with Type III dislocations. Most of the patients underwent surgery 7 (0–21) days after trauma. Follow-up was conducted at a median of 55 (29–90) months. In two patients, a dislocation of the buttons was observed during the follow-up period, both due to another direct trauma. One of them underwent revision surgery due to ongoing pain. He was treated with the Weaver and Dunn technique and therefore excluded from follow-up. The other patient did not express any symptoms at latest follow-up and was satisfied with the outcome, so revision was not deemed necessary in this case. We did not observe any other surgery related complications such as injuries of nerves and blood vessels, damage to the lung, or early implant failure.

### 3.1. Functional Results and Horizontal Instability

As for functional results, the patients scored an average Constant of 95 (±8.8), a UCLA of 31 (±4.9), and a DASH of 9.1 (±14.3) points. The VAS score was approximately at 0.9 (±0.1). A total of 36 (71%) patients reported ‘high satisfaction’ and 14 (27%) ‘moderate satisfaction’ with the treatment. Only one patient (2%) was not satisfied with the result and stated that she would have rather undergone conservative treatment instead. A persistent horizontal instability could neither be found intra- nor post-operatively in any of the cases. Table 2 illustrates an overview of the clinical outcome scores (Table 2).

### 3.2. Influencing Factors on Outcome

In our series, patient age at the time of injury had a significant influence on clinical outcome, with younger patients reaching better scores of the UCLA (*p* = 0.006) as well as the VAS (*p* = 0.004). DASH scores did not show any significant differences, yet a trend (*p* = 0.131) towards age as an influencing factor. A cut-off value could not be defined, since the sample size was too small for a valid statement.

In our series, the age of the patient at the time of injury had a significant influence on clinical outcome, with younger patients reaching better values in the UCLA score (*p* = 0.006) as well as in the VAS (*p* = 0.004). DASH scores did not show any significant difference, but also a trend (*p* = 0.131) towards age as an influencing factor. A cut-off value could not be defined, since the sample size was too small for a valid statement.

Another finding was the effect of the time to surgery on the clinical scores, where we showed that a longer time to surgery had a significant negative influence on all scores (*p* ≤ 0.001) except for the DASH (*p* = 0.180). We tried to identify a cut-off and examined a possible difference between the patients operated less than 10 days after trauma and the patients who had undergone surgery later. We observed better score values in absolute numbers for all of the collected data, yet only the Constant score showed significantly better results (*p* = 0.029) for the group of patients operated in under 10 days. Nevertheless, an increased CC distance (13.6 vs. 10.4 mm; *p* = 0.003) was shown for the patients operated after 10 days following injury.

The CC distance, as the radiological measure of surgical success, significantly affected the clinical outcome regarding the Constant (*p* = 0.007) and UCLA scores (*p* = 0.035) at follow-up, whereas the VAS (*p* = 0.068) and DASH scores (*p* = 0.655) were not affected.

The primary grade of injury according to Rockwood had no significant influence on the clinical outcome.

### 3.3. Radiological Results

According to Taft et al. [38] 30 patients (59%) had no signs of reduction loss. Nine patients (18%) presented with a slight loss of reduction. In 11 patients (22%) a partial loss and in one patient (2%) a total loss of reduction was detected. No significant difference regarding clinical scores could be shown between the groups. In absolute numbers, we observed an average of 2 mm reduction loss (9.1 vs. 11.2 mm) over the whole patient group.

Only four (8%) patients developed a radiologically visible AC joint arthritis at follow-up. They mostly correlated with a positive horizontal adduction test (*p* = 0.028) and a lower VAS score (*p* = 0.03).

Calcification of the CC ligaments could be observed in 19 cases (37%), but no correlation could be found between the radiological findings and the clinical outcome (*p* = 0.275).

In the intraoperative dynamic transaxillary views, horizontal movement of more than 50% shaft width could not be observed in any of the patients. An overview of the radiological results can be found in Table 3 (Table 3).

## 4. Discussion

Acute Rockwood IV–VI injuries should be managed surgically [10]. Arthroscopic assisted or minimally invasive reconstruction such as MINAR^®^ or TightRope^®^ fixation are superior to static techniques such as the Bosworth screw and K-wire transfixation [3], presenting with similar or even better results compared to other techniques [39,40,41].

Summarizing, we observed good clinical outcome results (Constant, VAS, UCLA, and DASH scores) which are comparable to other MINAR^®^ treated series such as by Rosslenbroich et al. [42] or Petersen et al. [43]. Table 4 provides a comparison of clinical results of other less rigid fixation methods (Table 4).

Younger patients reached significantly better test results compared to older ones, which corresponds to current literature [42]. A greater time interval to surgery was also associated with reduced test results, which has already been described [21]. Our patients benefitted from early surgery in under 10 days following trauma. This is an important fact, especially in treating patients with Rockwood III injuries since the ISAKOS recommends a reevaluation of conservative to surgical treatment 3 weeks after injury. In our own approach we suggest an immobilization in a sling bandage for 1 week and a subsequent clinical re-evaluation. If painful flexion and abduction persists and a horizontal instability is evident and confirmed via Alexander view, we recommend the surgical pathway. If the patient is able to flex without pain, is elderly, and has no extensive physical demand or does little overhead work, we recommend conservative treatment. Nevertheless, it has to be taken into account that the sample size was too small to give a definitive statement. Additional studies are absolutely necessary to further scrutinize these findings.

Open surgical methods for the treatment of AC joint injuries are sometimes criticized for the missing opportunity of simultaneous treatment of concomitant intra-articular pathologies, especially SLAP lesions [18,19,20]. Overall, rates differ between 15% [19] and 40% [18], mostly caused by degenerative changes, with an increasing number with advancing age [18].

Persisting dynamic horizontal instability is currently an intensively debated topic. Inferior clinical results due to solitary CC stabilizing measures have been discussed [14,49]. In our own approach we chose a skin incision, which allowed direct visualization of the AC joint, sutured the ruptured AC joint capsule and delta fascia. We suspect that the treatment of the soft tissue and the V-shaped array of the MINAR^®^ are the reasons why we could not detect any remaining horizontal instability neither intraoperatively nor at follow-up.

In our series, we observed a significant loss of reduction in over 20% of our patients during follow-up, but only a trend towards poorer results. These findings correspond to current literature where similar rates of clinically silent reduction loss could be shown [17,45,47]. During follow-up, a greater CC distance was observed in patients operated on more than 10 days after trauma, negatively influencing the Constant and UCLA scores. These findings support our suggestion for an early as possible surgical intervention in patients with an indication for surgery.

The rate of CC ligament calcification lies around the described rate in literature [50,51]. The pathological significance is still discussed, but asymptomatic findings are common. Bone fragments released during drilling are discussed as a cause [51]. In accordance to literature our findings did not reveal any negative influence on the clinical outcome. Only in two cases did we observe dislocation of the buttons. Both were caused by another unrelated direct trauma. In our opinion, the lack of button dislocation which is described at around 10% [42] can be attributed to the V-shaped array with a total of three Flip Tacks, which guarantee better stability than a single system. We did not find any other surgical complications such as nerve damage, injury of blood vessels, or infection, but four patients developed a painful osteoarthritis of the AC joint. It has to be assumed that some kind of microinstability remained in these cases, which at least partially contributed to the development of the osteoarthritis, even if we were not able to quantify it with our measurements.

A weakness of this trial is its retrospective study design and the heterogenous and only mid-term follow-up. Additionally, the lack of routine Alexander view X-rays and standardized measurements for persistent horizontal instability, especially at follow-up, is a point for criticism. At this point it has to be stated that our patients were operated on well before Zumstein published the standardized X-ray measurements. Furthermore, the large age distribution is a point of criticism. However, to our knowledge it is the first analysis of the triple-button MINAR^®^ system in current literature and the number of included patients is rather high. Furthermore, measurements on radiographs as well as the clinical examination and calculation of the scores were made by the same person, who did not perform the surgery and was not employed at the hospital.

## 5. Conclusions

In the treatment of acute AC joint dislocation, we can show very good clinical outcomes using the MINAR^®^ system, which are comparable to other less rigid surgical methods. In comparison to other surgical options the MINAR^®^ technique requires less surgical know-how and skill. If there is an indication for surgical treatment, patients benefit from an early operation. Refixation of the AC joint capsule, as well as suturing the ruptured delta fascia is possible over the same skin incision to augment horizontal stability. Further biomechanical and clinical studies are necessary to prove improved horizontal instability after a V-shaped button array and soft tissue repair.

## Figures and Tables

**Figure 1 jcm-08-01683-f001:**
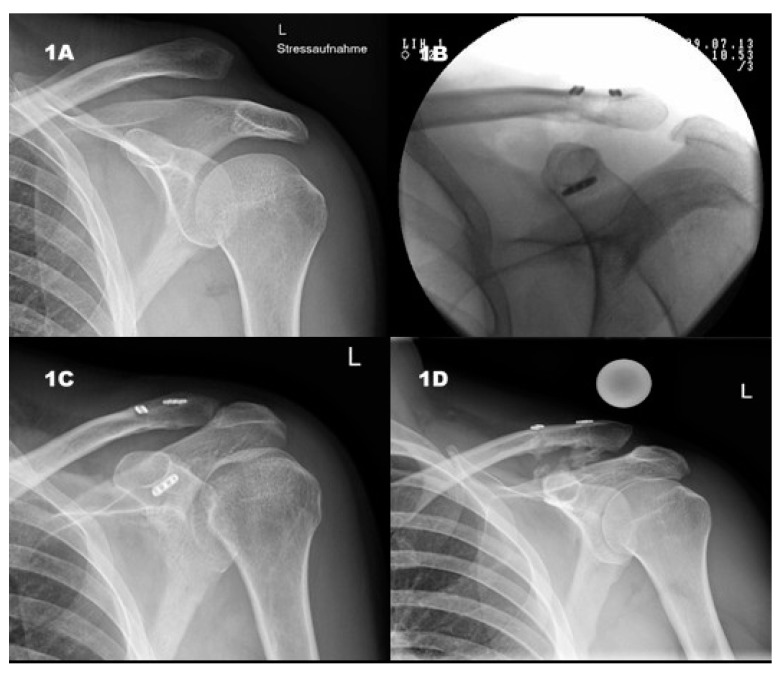
(**1A**) Preoperative, Rockwood V lesion, intraoperative, (**1B**) intraoperative, V-shaped array; (**1C**) postoperative, anatomical reduction; (**1D**) follow-up, stress view, slight loss of reduction, CC-ligament calcification.

**Figure 2 jcm-08-01683-f002:**
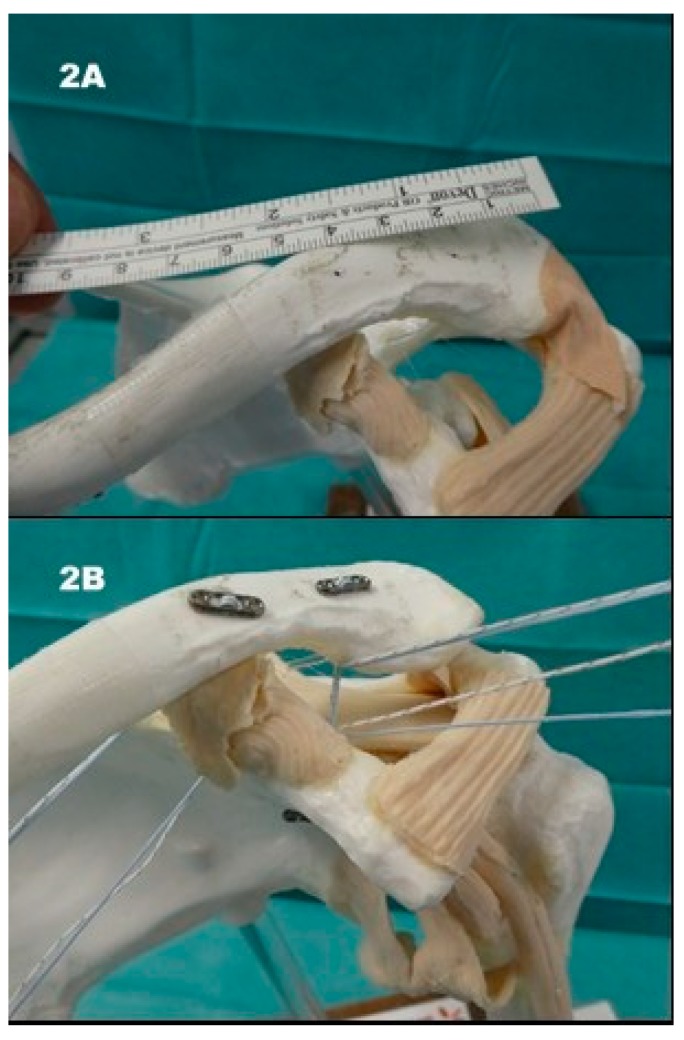
MINAR model: (**2A**) at least 2 cm distance between drill holes at the lateral clavicle; (**2B**) V-shaped array of the triple-button system. Knotting between coracoid and clavicle.

**Figure 3 jcm-08-01683-f003:**
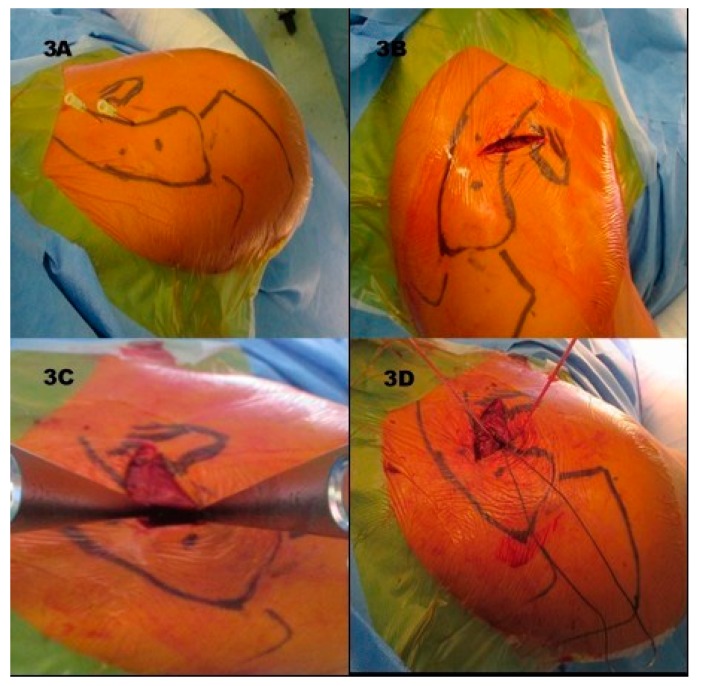
(**3A**) Landmarks, dots representing drill holes at clavicle, needles coracoid base; (**3B**) skin incision over coracoid base and (**3C**) Hohmann hooks at the coracoid base; (**3D**) acromioclavicular (AC) capsule sutures on the right sight, FiberWire sutures in cranial direction.

**Table 1 jcm-08-01683-t001:** Patient characteristics. Y = years, m = male, f = female, d = days, mo = months; * Data reported as median (range).

Patient Characteristics	
**Age (y)**	43 (19–63) *
Sex (m/f)	62/3
Rockwood Type	Type III *n* = 8 (16%)
	Type IV *n* = 11 (21%)
	Type V *n* = 32 (63%)
Time to surgery (d)	7 (0–21) *
Follow up (mo)	55 (29–90) *
Lost to FU	14 (21.5%)
Overhead work	21/65 (32%)
Previous shoulder surgery	none
Smoking	8/65 (12%)
Sports (main)	Cycling *n* = 13 (20%)
	Hiking *n* = 11 (17%)
	Soccer *n* = 7 (10%)
	Running *n* = 14 (22%)
	Other *n* = 18 (27%)
	None *n* = 2 (4%)
Comorbidities	Art. Hypertension *n* = 1
	Hypothyreoidism *n* = 2
	Hypercholesterolemia *n* = 1

**Table 2 jcm-08-01683-t002:** Functional outcome (FU). The functional outcome at the time of follow up is pictured here.

Functional Outcome (FU)		
Score	Value	Reference Value
Constant	95 (±8.8)	0–100
DASH	9.1 (±14.3)	0–10
UCLA	31 (±4.9)	100–0
VAS	0.9 (±0.1)	1–0
Satisfaction (n/%)	high *n* = 36 (71%)	
	moderate *n* = 14 (27%)	
	no *n* = 1 (2%)	

Constant score: 0 = worst–100 = best; DASH score: 0 = worst–10 = best; UCLA score: 100 = worst–0 = best; VAS score: 1 = worst–0 = best; all score values are displayed as arithmetic mean and standard deviation. DASH: Disabilities of Arm, Shoulder and Hand UCLA: University of California Los Angeles Shoulder score VAS: Visual Analogue Scale.

**Table 3 jcm-08-01683-t003:** Radiological outcome. Coracoclavicular (CC) and Acromioclavicular (AC) distances are displayed in millimeters (mm). Not available (n.a.). The values are given as arithmetic mean and standard deviation.

Radiological Outcome			
	prae OP	post OP	Follow-Up
CC distance (mm)	23.00 (±5.49)	9.10 (±3.11)	11.25 (±3.14)
AC distance (mm)	11.25 (±4.47)	4.25 (±2.14)	5.82 (±2.78)
AC arthritis (n/%)	n.a.	n.a.	4/51 (8%)
CC ligament calcification (n/%)	n.a.	n.a.	19/51 (37%)

**Table 4 jcm-08-01683-t004:** Data is reported as mean ± standard deviation (range). N.r., data not raised; d, days; m, months.

Functional Outcome after Nonrigid AC Joint Repair								
Study	Technique	Number of Patients	Patient Age (y)	Interval Trauma to Surgery (d)	Follow Up (m)	Constant Score	DASH	UCLA
Wang et al. (2018) [1]	allogenic Tendon Graft	8	49 (23–72) *	<21	29.8 (25–43)	94.4 (86–100)	n.r.	33.5 (30–35)
Yin et al. (2018) [44]	Tendon Graft (conjoined Tendon Autograft)	25	46.28 (20–68)	4.86 ± 1.17	19.92 ± 2.92	89.56 ± 2.80	n.r.	n.r
Hann et al. (2017) [17]	Double TightRope + AC cerclage	59	43.3 (24.4–56) *	<21	26.4 (20.3–61) *	90 (33–100) *	n.r.	n.r
Vulliet et al. (2017) [45]	Double Tight Rope	22	38.8 ± 8.7	3 ± 1.9	27.7 ± 8.3	94.3 ± 4.4	2.0 ± 2.6	n.r
Beris et al. (2013) [46]	Single TightRope	12	27.5 (19–39)	5 (2–14)	18.3 (12–30)	94.8 (84–100)	0.25 (0-3)	n.r.
Tiefenboeck et al. (2018) [47]	LARS	47	37.3 (17–65) *	8 (<14)	90 (25–159) *	93 (5–100) *	2.6 (0–31) *	35 (20–35) *
Lu et al. (2013) [48]	LARS	24	31 (21–45) *	<42	36 (6–60) *	94.5 ± 9.3	n.r.	n.r.
Rosslenbroich et al. (2015) [42]	MINAR	83	39 (17–80) *	6 (0–22)	39 (12–78)	94.7 (61–100)	n.r.	n.r.
Breuer et al. (2018)	MINAR	51	43 (19–63)	7 (0–21)	55 (29–90) *	95 ±8.8	9.1 ± 14.3	31 ± 4.9

* Data reported as median (range).

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
