# Peer review of "Minimally Invasive AC Joint Reconstruction System (MINAR®) in Modified Triple-Button Technique for the Treatment of Acute AC Joint Dislocation"

_jcm, 2019, doi:10.3390/jcm8101683_

Round 1

Reviewer 1 Report

This is a paper with some important findings suggesting age and time of surgical intervention after injury correlate with outcomes, however clinical significance of those differences is hard to ascertain. Presenting some of this data in table form so the reader can better elucidate that is recommended.  

There is no control group in this project for comparison, limiting its statistical soundness, but overall a well done and presented retrospective study. 

Author Response

This is a paper with some important findings suggesting age and time of surgical intervention after injury correlate with outcomes, however clinical significance of those differences is hard to ascertain. Presenting some of this data in table form so the reader can better elucidate that is recommended.  

Thank you for your kind words and thank you again for your valuable input. We added a table with patient characteristics in the experimental section. Furthermore, we added two tables in the results section which give an overview of the functional and radiological outcomes. I hope these changes will make it easier to comprehend and assess our findings.

There is no control group in this project for comparison, limiting its statistical soundness, but overall a well done and presented retrospective study. 

Thank you very much for your positive review. It is our goal for the future to carry out  comparative studies with patient series treated conservatively as well as to compare the MINAR® system to other flexible fixation methods. A study comparing the MINAR system to TightRope® fixation is already in the making and will soon be submitted.

Reviewer 2 Report

Dear Authors,

I realize that authors have many journals to consider when they want to publish their work, so I appreciate your interest in Journal of Clinical Medicine. It is apparent that you have put a great deal of effort into this project and I want to applaud your efforts.

I am very sorry to be able to write in a more positive way but It may be that you would like to consider re-submitting it after major revision, in which case I hope that the comments from my review may help you to revise it before re-submitting. These comments are given below.

It is evident that you have put a great deal of effort into this project and I want to praise your efforts, but as noted in the instructions of trial design study, feature focuses on the use and interpretation of therapeutic intervention data related to a case scenario relevant to orthopaedic surgery, in most instances these data will emphasize how information from trials can impact medical or physical therapy management of this kind of patients. Unfortunately, the actual contribution from your paper to the surgery for AC joint dislocation literature is not totally clear or strong. The manuscript as currently written suggests that it might be suitable for sharing information about this particular clinical condition, but the paper you reported is not representative to state with certainty your conclusions and it need of major revision.

In outcome and follow-up: None information about other symptoms are reported in these cases for example: kinesofobia, reduction in partecipation levels and none other strong outcomes were been mesured in these cases: i.e. catastrophizing, depression profile, anxiety, and no evaluation scales was used for these PROM. 

Major Revision:

Experimental Section: describe with the name, the study design used.

Experimental Section - Patients:

line 60 (A consecutive series of 65 patients treated): add please a table (after, you remember to ri-numerated the tables in the text) with characteristics of patients: age, gender, time passed from injury, BMI, work o sport activity, smoke or not, other comorbidities, other previous surgery, etc, etc

line 61 (time from injury to surgery <21 days): add please median and standard deviation of 65 patients. Understanding the time between injury and surgery could help future research and create new, more timely intervention protocols;

line 66 (the local ethics committee 66 approved of the trial). please provide this number. for your research here and in the abstract section.  If you don't written cannot proceed with publication.

line 67 (At the time of follow-up 51 patients voluntarily participated (21.5% Lost to FU). please provide to insert the table (after, you remember to ri-numerated the tables in the text) for description the time of follow-up, how many follow-up you have made, and, insert the variations of patient reported outcomes measure (PROM).

Experimental Section - Clinical examination and scores:

line 82-84 Please insert the name of each tests used in clinical examination of patients: i.e. for the rotator-cuff (RC) tests and impingement tests, exist much different tests;

line 87-92 Please insert the references for each outcomes measure scale used

Results section:

line 132-190 aggregate the data in a one or two table. This action will facilitate the lecture of paper and understanding of change od PROM in follow-up time.

line 137 insert the time of follow up; insert the data in order to one or more follow up that you made in the time.

Minor Revision

Introduction section: 

In this Section missing many: references in different sentences:

line 32  missing references after "injuries";

line 40 missing reference after "(IIIb)";

line 41  missing references after "injuries";

line 42  missing references after "suggested";

line 48 missing references after "standard".

Abstract and Introduction section: 

In abstract (line 13) and in line 31 of introduction section explode the AC that it's appear for the first time;

In abstract (line 20) and in line 51 of introduction section explode the CC that it's appear for the first time;

In reference section: the references appear with two numbers of numeration

Round 2

Reviewer 2 Report

Dear Authors

It is evident that you have put a great deal of effort into this review process.The manuscript has been significantly improved. I am very happy to be able to write in this positive way.

Best Regards